# The Role of Anti-Factor Xa Activity in the Management of Ecchymosis in Patients Receiving Rivaroxaban after Total Knee Arthroplasty

**DOI:** 10.3390/jcm12031161

**Published:** 2023-02-01

**Authors:** Han Wang, Jiacheng Liu, Xiaofei Lai, Xinyu Li, Wei Huang

**Affiliations:** 1Department of Orthopedics, Zhangzhou Affiliated Hospital of Fujian Medical University, Zhangzhou 363000, China; 2Department of Orthopedics, The First Affiliated Hospital of Chongqing Medical University, Orthopedic Laboratory of Chongqing Medical University, Chongqing 400016, China; 3Department of Laboratory, The First Affiliated Hospital of Chongqing Medical University, Orthopedic Laboratory of Chongqing Medical University, Chongqing 400016, China; 4Department of Pharmacy, The First Affiliated Hospital of Chongqing Medical University, Orthopedic Laboratory of Chongqing Medical University, Chongqing 400016, China

**Keywords:** anti-factor Xa activity, anti-coagulation, rivaroxaban, ecchymosis, blood loss, total knee arthroplasty

## Abstract

This study aims to evaluate the efficacy of anti-factor Xa activity (aFXa) in predicting ecchymosis after total knee arthroplasty (TKA). One hundred and two unilateral primary TKA patients were recruited consecutively in this prospective observational study. Participants received rivaroxaban (10 mg p.o. qd) from postoperative day 1 (POD1) to POD35 and were divided into a non-ecchymosis group (group A) and an ecchymosis group (group B). AFXa was assessed as the primary outcome on POD1 and POD3. Prothrombin time (PT), activated partial thromboplastin time (APTT) and thromboelastography (TEG) were recorded both preoperatively and postoperatively (on POD1 and POD3). Other outcomes, including venous thromboembolism (VTE), blood loss and wound complications were also collected and compared. As a result, 27.5% of the participants (n = 28) were allocated into group B. Demographic data were comparable between the two groups. The aFXa levels in group B were significantly higher than those in group A on POD1 and POD3, and the aFXa level was assessed as an independent risk factor for ecchymosis. The cut-off value of aFXa was determined to be 121.38 ng/mL at maximal Youden index, associated with area under the receiver operating characteristics curve of 0.67. Group B experienced significantly more blood loss and wound complications than group A. No statistical difference was detected regarding PT, APTT and TEG parameters. AFXa is a promising parameter to predict ecchymosis after TKA. Patients with aFXa > 121.38 ng/mL should be considered as high-risk population for postoperative ecchymosis and may require intense monitoring or dosage modification of anticoagulants.

## 1. Introduction

Total knee arthroplasty (TKA) is an effective surgical intervention for end-stage knee diseases [1]. Patients receiving TKA are stratified in a high-risk group of venous thromboembolism (VTE) according to the Caprini Score for thrombosis risk assessment [2,3]. Fortunately, due to the wide application of anticoagulants according to the guidelines, the rate of VTE following TKA has dropped significantly in the last few years [2,3,4]. However, it is noticeable that more postoperative bleeding complications (such as ecchymosis, wound and gastrointestinal bleeding) were also observed with the universal use of anticoagulants, and it has been considered a potential adverse result of the drug-induced coagulation dysfunction [5,6,7,8,9,10].

Ecchymosis is a common bleeding complication related to the postoperative coagulopathy after TKA, of which incidence was reportedly up to 33% [5,11]. Ecchymosis is bleeding in the third space under subcutaneous tissue, which can aggravate the swelling and inflammation around the surgical site, slow down the postoperative rehabilitation process and increase a patient’s anxiety after surgery [7,12]. Nevertheless, it remains challenging to minimize the VTE rate without increased bleeding risk. Meanwhile, according to the 9th Edition of *American College of Chest Physicians Evidence-Based Clinical Practice Guidelines* (ACCP-9), the duration of thromboprophylaxis in patients undergoing TKA was suggested to be extended for up to 35 days postoperatively [4]. As such, when TKA patients develop ecchymosis during the 35-day anti-coagulation therapy, orthopedists are always in a dilemma whether to continue or discontinue the anti-coagulation treatment. Therefore, it is important to find an effective measurement to help with these clinical decisions.

Chromogenic anti-factor Xa activity (aFXa) assay is a widely used measurement to evaluate the intensity of anticoagulation and drug quantification in patients receiving direct anti-Xa inhibitors [13,14,15]. Several studies have demonstrated that aFXa level had superior linearity with the plasma concentration of rivaroxaban [16,17,18,19,20]. When compared with prothrombin time (PT), activated partial thromboplastin time (APTT) and thromboelastography (TEG), aFXa could monitor the plasma concentration and pharmacokinetics of rivaroxaban more accurately [21,22,23,24,25]. Therefore, aFXa may be a promising parameter to monitor bleeding complications. Nonetheless, the role of aFXa in evaluating the effectiveness of anticoagulants and predicting postoperative bleeding events (such as ecchymosis) after TKA remains uncertain.

Hence, we conducted this prospective observation trial, aiming to (1) assess the efficacy of aFXa in predicting ecchymosis after TKA; (2) identify risk factors contributing to ecchymosis following TKA.

## 2. Materials and Methods

### 2.1. Study Design

This is a prospective observational study approved by the institutional ethics committee. Knee osteoarthritis patients admitted to our center consecutively between February 2019 to August 2019 were assessed for enrollment. Informed consent was obtained from all enrolled participants before study inclusion. All surgeries were conducted by the same surgical team under consistent anesthetic, surgical and perioperative management protocols.

### 2.2. Participants

Patients were considered eligible if they (1) were 18 years of age or older; (2) underwent unilateral primary TKA; (3) received rivaroxaban for VTE prophylaxis after TKA; (4) provided informed consent.

Patients were excluded if they (1) underwent bilateral TKA or revision TKA; (2) were diagnosed with VTE or received vascular surgery in the last 6 months before admission; (3) were presented with comorbidities of severe hepatic dysfunction (Child-pugh C), renal diseases (creatinine clearance < 30 mL per minute) or other severe systemic diseases; (4) were pregnant or breast-feeding; (5) had any other contraindications of rivaroxaban.

The flow chart of the study is illustrated in Figure 1.

### 2.3. Perioperative Management

All TKAs were performed under general anesthesia through the standard medial parapatellar quadriceps-splitting approach with patients in supine position. All enrolled patients received the same perioperative care. Tourniquet with a fixed pressure of 30 kPa was routinely used during the operation. Moreover, 1.5 g of cefuroxime sodium was administered intravenously 30 min before the incision, and then the same dose was given every 12 h and discontinued 48 h after surgery for perioperative infection prophylaxis. Meanwhile, 1.5 g of tranexamic acid was given intravenously 30 min prior to skin incision, and another dose of 1.0 g was administered intra-articularly before wound closure. No postoperative tranexamic acid was given, and no surgical drains were used postoperatively in our routine TKA procedure.

Next, 10 mg of rivaroxaban (Xarelto, Bayer Schering Pharma, Berlin, Germany) was given once daily at 8 A.M. from post-operative day 1 (POD1) to POD35 to prevent VTE according to the ACCP-9 guideline [4]. Pneumatic compression pump was applied to all TKA patients postoperatively to prevent VTE mechanically. Furthermore, all TKA patients received therapeutic exercise guided by physical therapists from POD1 to the day of discharge according to the rehabilitation protocol. All patients were scanned for deep venous thrombosis (DVT) using color doppler ultrasonography for the veins of lower extremities 1 week after surgery.

Ecchymosis was defined as subcutaneous hemorrhagic lesions, larger than >3 mm, forming a flat, rounded or irregular, blue or purplish patch in this study [26,27]. Patients were monitored daily from POD1 to the discharge day by the researchers for the occurrence of ecchymosis and then were divided into ecchymosis group and non-ecchymosis group later in the analyses. In the ecchymosis group, the location and area of the ecchymosis were also recorded. The area of a patient’s single hand was regarded as 1% of the whole-body surface area [28].

### 2.4. Outcomes and Data Collection

Demographic data (including age, gender, height and weight) were collected once patients were recruited. The serum peak concentration of aFXa was tested on POD1 and POD3 at 3 h after the drug intake, and it was recorded as the primary outcome. All the chromogenic aFXa assays were carried out using the STA^®^—Liquid Anti-Xa (Diagnostica Stago, Parsippany, NJ, USA) with hybrid anti-factor Xa calibration methods. Secondary outcomes, including PT, APTT, TEG, serum creatinine, glomerular filtration rate (GFR) and hematocrit (Hct) were assessed and collected both pre- and post-operatively. Other outcomes (such as thromboembolism, bleeding and incision complications) were also monitored daily during the hospitalization.

Total blood loss (TBL) and hidden blood loss (HBL) were estimated based on the Gross equation [29]. The specific formula was as follows: TBL = patient blood volume (PBV)^2^ × (pre Hct − post Hct)/(pre Hct + post Hct), where the PBV = k1 × [height (m)]^3^ + k2 × weight (kg) + k3. For males, k1 = 0.3669, k2 = 0.0329 and k3 = 0.6041; for females, k1 = 0.3561, k2 = 0.03308, and k3 = 0.1833. Meanwhile, HBL = TBL − intraoperative blood loss, where the intraoperative blood loss was acquired through the surgical record. Both the TBL and HBL on POD1 and POD3 were calculated.

### 2.5. Statistical Analysis

Statistical analysis was performed using the SPSS software for Windows, version 20.0 (SPSS Inc., Chicago, IL, USA) and MedCalc Statistical Software (version 16.8.4, Ostend, Belgium). Demographic data between the 2 groups were compared using the Pearson χ^2^ test (for variables such as gender) and independent *t*-test (for variables includin age and body mass index). The performance of aFXa value as an estimated test in the postoperative period was assessed through receiver operating characteristics (ROC) curve analysis. In this analysis, the area under the curve (AUC) is proportional to the usefulness of a test to correctly discriminate between subjects with and without ecchymosis, with AUC = 1.0 in an ideal test. The sensitivity, specificity, positive/negative predictive value and the positive/negative likelihood ratio were also calculated. The Youden’s J statistic was used to select sets of the optimal cut-off values, with the final decision based on the clinical availability. Statistical significance was defined as a *p* value of <0.05. For most values, 95% confidence intervals (95% CI) were computed with Fisher’s Exact test.

## 3. Results

### 3.1. Participant Flow and Baseline Characteristics

A total of 131 TKA patients were screened during the study period (Figure 1). However, 29 of them were excluded: 12 received preoperative anti-coagulation, 7 declined to participate, 5 received revision TKA, 3 received bilateral TKA and 2 were combined with severe hepatic or renal dysfunction preoperatively. Finally, 102 patients were recruited in this study. The 74 patients (72.5%) without ecchymosis and the other 28 patients (27.5%) developed various degrees of ecchymosis after TKA and were divided into a non-ecchymosis group (group A) and an ecchymosis group (group B), respectively. Most ecchymosis appeared on POD3 and were mainly located around the surgical site. The mean area of ecchymosis was 2.59 ± 2.27% of body surface. No significant difference was observed about the baseline characteristics (such as age, body mass index (BMI), hepatic and renal function) between the two groups (Table 1).

### 3.2. Anti-Factor Xa Activity and Routine Coagulation Parameters

The aFXa levels of group B were significantly higher than those of group A, both on POD1 (*p* = 0.01) and POD3 (*p* < 0.01) (Table 2). ROC curves were constructed to show the relationship between a true positive and a false positive when using postoperative aFXa to predict ecchymosis. The cut-off value of aFXa on POD3 was determined to be 121.38 ng/mL at maximal Youden index, associated with an area under the ROC curve of 0.67 (95% CI 0.57–0.76) (Figure 2). No statistical difference was found between the two groups regarding perioperative values of PT, APTT and the variables of TEG. (Table 2 and Table 3). The sensitivity, specificity, positive predictive value (PPV), negative predictive value (NPV), positive likelihood ratio (LR+) and negative likelihood ratio (LR−) at postoperative aFXa cut-off value of 121.38 ng/mL are shown in Table 4.

### 3.3. Blood Loss and Other Complications

There was no significant difference between the two groups regarding operation time and intraoperative blood loss (Table 5). Group B suffered significantly more TBL (302.09 ± 87.86 mL vs. 253.85 ± 89.05 mL, *p* = 0.02) and HBL (273.16 ± 92.19 mL vs. 218.64 ± 93.53 mL, *p* = 0.01) than group A (Table 5). Five cases of asymptomatic distal DVT after surgery were detected by the color doppler ultrasonography of the lower extremities: three in group A and two in group B. There was no significant difference in the incidence of DVT between the two groups (4.05% vs. 7.14%, *p* = 0.52) (Table 6). More wound complications were observed in group B than in group A (4.05% vs. 25.00%, OR = 7.89, 95% CI 1.87–32.21, *p* < 0.01) (Table 6).

### 3.4. Risk Factors for Ecchymosis after Total Knee Arthroplasty

Perioperative clinically relevant factors with a *p* value < 0.05 in the univariate model are included in the multivariate regression analysis (Table 7). The aFXa levels on POD1 and POD3 were found to be independent risk factors, with adjusted odds ratio of 1.015 (95% CI 1.001–1.028, *p* = 0.04) and 1.015 (95% CI 1.003–1.028, *p* = 0.02), respectively.

## 4. Discussion

The principal findings of this study are as follows: (1) aFXa is a promising parameter in predicting ecchymosis following TKA while taking rivaroxaban for DVT prophylaxis; (2) aFXa is an independent risk factor of ecchymosis after TKA; (3) TKA patients with aFXa level > 121.38 ng/mL should be considered as a high-risk group for postoperative ecchymosis, indicating a higher possibility of wound complications and a dosage adjustment if necessary.

As an oral direct factor Xa inhibitor, rivaroxaban is a recommended anticoagulant in routine TKA process, which has proven efficacy in preventing VTE during the perioperative period [4,10,12]. However, several clinical studies revealed that rivaroxaban can contribute to a higher risk of bleeding complications when compared with aspirin and low-molecular-weight heparin [30,31,32]. Since bleeding events (such as ecchymosis) would impede the postoperative rehabilitation process and increase wound complication risk, researchers are trying to find a practical measurement to monitor the efficacy and safety of anticoagulants following TKA procedures.

Routine coagulation tests, such as PT, APTT and TEG, have been widely used in various surgical areas [5,33]. However, these parameters seemed to provide limited efficacy in monitoring and predicting ecchymosis. PT and APTT are the most frequently used clot-based assays, which are inexpensive and rapid, but they are less reliable in reflecting anti-factor Xa inhibitor exposure when compared with aFXa [14]. Similarly, no significant difference was detected in PT and APTT between the two groups in this study. In this context, it seems that chromogenic aFXa assay, which was applied originally for the determination of the rivaroxaban plasma level, is a promising measurement for monitoring bleeding events in a variety of scenarios requiring anticoagulation [34,35,36]. Since the level of aFXa was proved to have superior linear relation with the anti-factor Xa inhibitor plasma concentration [14], we hypothesized that patients with ecchymosis may have a higher aFXa level than those without [16,17,18,19,20]. Consistent with our hypothesis, group B had significantly higher levels of aFXa both on POD1 and POD3 than group A in this study. This may indicate that the routine anticoagulation therapy was not suitable for all TKA patients, and the recommended dosage was perhaps excessive for patients with ecchymosis after TKA. Given that most of the ecchymosis appeared 2–3 days after surgery [5,37,38] and the level of aFXa in patients with ecchymosis was already significantly higher than those without on POD1, aFXa may be a promising parameter to predict ecchymosis before its onset and help orthopedists to modify the dosage timely.

Based on the findings above, we further conducted a regression analysis of potential risk factors for ecchymosis after TKA and found that the aFXa level was an independent risk factor for postoperative ecchymosis (Table 7). In addition, we identified two specific cut-off values to further evaluate the effectiveness of aFXa in predicting ecchymosis after TKA (Table 4). According to the obtained results from this study, patients with aFXa levels over 121.38 ng/mL should be considered as high-risk for postoperative ecchymosis. The accuracy of this cut-off value was also validated with the ROC curve analysis, yielding an area under the curve of 0.67. Furthermore, according to our analysis, postoperative aFXa > 180.12 ng/mL is strongly suggestive of the onset of ecchymosis (PPV: 100%). The two cut-off values may help orthopedists identify high-risk patients for ecchymosis and modify the postoperative anticoagulation protocol timely, thus reducing the risk of bleeding complications.

In a previously published work, Wang et al. reported that the perioperative blood loss of patients with ecchymosis after TKA was more than those without [5]. Consistently, according to our results, patients with ecchymosis suffered significantly more blood loss and wound complications than those without (Table 5 and Table 6). Hence, timely identification of high-risk patients and the prevention of postoperative ecchymosis are of vital importance to reduce these risks, and the perioperative monitoring of aFXa level may contribute to the management and prediction of ecchymosis after TKA. More importantly, assessment of aFXa level may help orthopedists with individualized anticoagulation management in special populations, such as the elderl, or patients with hepatic or renal dysfunction, achieving a balance between the risk of thrombotic and bleeding events.

There are some limitations to the present study. Firstly, the small sample size of this study may result in the underestimation of some variables, such as older age, HBL, albumin and hemoglobin level, which were already reported as the risk factors of ecchymosis [5,39,40]. Secondly, since chromogenic aFXa assay was performed only on POD1 and POD3 due to some objective reasons, we failed to provide the dynamic monitoring of aFXa levels in this study. Thirdly, as the participants were recruited from one single medical center, the research findings may not necessarily generalize to a broader population. Fourthly, further prospective studies monitoring more coagulation parameters (such as platelet and coagulation factors) are needed to provide a more comprehensive view of the relationship of anti-factor Xa inhibitors, aFXa and coagulation function.

## 5. Conclusions

To conclude, aFXa is a promising parameter in predicting ecchymosis among patients taking rivaroxaban after TKA, assisting in identifying high-risk patients and guiding the adjustment of anticoagulation dosage timely. It is also an independent risk factor of ecchymosis after TKA. Patients with an aFXa level > 121.38 ng/mL in clinic could be considered as a high-risk group for ecchymosis and may require dosage modification of anticoagulants.

## Figures and Tables

**Figure 1 jcm-12-01161-f001:**
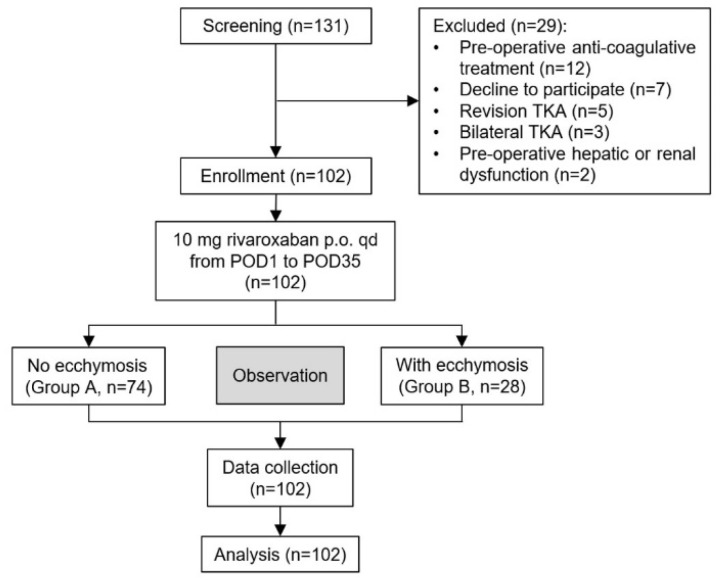
The flow chart of study. TKA, total knee arthroplasty; POD, post-operative day.

**Figure 2 jcm-12-01161-f002:**
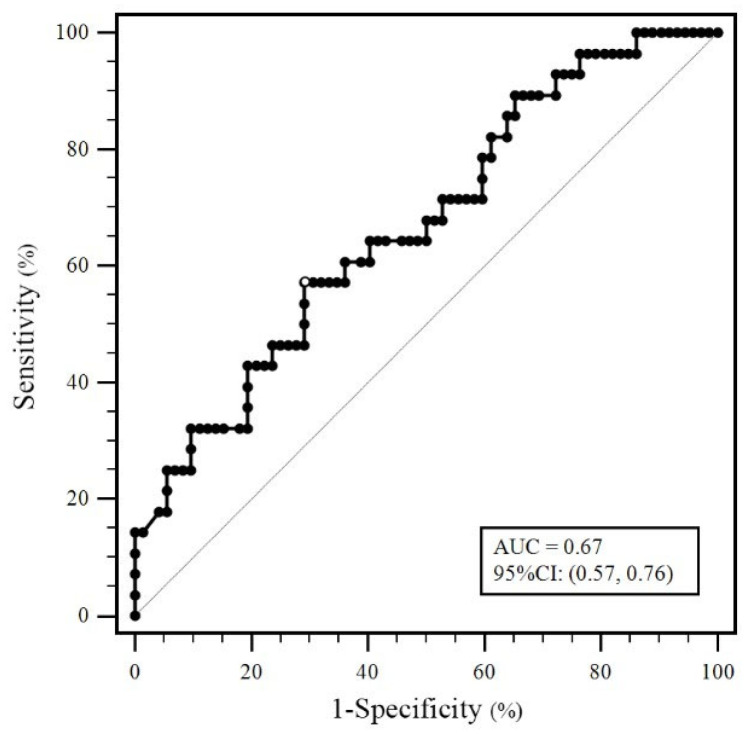
ROC curve examining the correlation between anti-factor Xa activity and ecchymosis.

**Table 1 jcm-12-01161-t001:** The baseline characteristics of enrolled patients.

	Group A (n = 74)	Group B (n = 28)	*p* Value
Age (year)	68.91 ± 9.82	69.39 ± 5.82	0.77
Gender (male/female)	9/65	2/26	0.47
BMI (kg/m^2^)	24.90 ± 2.90	25.87 ± 3.27	0.15
GFR (ml/min)	88.23 ± 19.87	86.37 ± 14.82	0.65
Scr (μmol/L)	69.47 ± 23.29	65.36 ± 15.80	0.39
ALT (U/L)	17.18 ± 13.43	16.57 ± 8.17	0.82
AST(U/L)	21.50 ± 16.47	19.00 ± 7.33	0.44

BMI, body mass index; GFR, glomerular filtration rate; Scr, serum creatinine; ALT, alanine aminotransferase; AST, aspartate amino transferase.

**Table 2 jcm-12-01161-t002:** The anti-factor Xa activity and routine coagulation parameters of enrolled patients.

	Group A (n = 74)	Group B (n = 28)	*p* Value
aFXa (ng/mL)			
POD1	73.75 ± 42.99	96.88 ± 38.19	0.01 *
POD3	100.15 ± 36.20	132.95 ± 54.76	<0.01 *
PT (s)			
Pre-OP	12.95 ± 0.62	12.98 ± 0.91	0.89
POD1	13.66 ± 1.67	13.53 ± 1.55	0.73
POD3	13.45 ± 1.44	13.58 ± 1.83	0.71
APTT (s)			
Pre-OP	36.04 ± 3.43	35.11 ± 4.09	0.25
POD1	34.78 ± 3.49	33.98 ± 5.33	0.38
POD3	36.16 ± 3.07	35.37 ± 4.59	0.32

aFXa, anti-factor Xa activity; POD, post-operative day; Pre-OP, pre-operation; PT, prothrombin time; APTT, activated partial thromboplastin time; *, statistically significant difference in mean values between groups.

**Table 3 jcm-12-01161-t003:** Specific values of TEG variables of enrolled patients.

	Group A (n = 74)	Group B (n = 28)	*p* Value
Pre-OP			
R (min)	5.40 ± 0.87	5.42 ± 0.97	0.93
K (min)	1.44 ± 0.40	1.48 ± 0.39	0.68
α-angle (°)	67.62 ± 7.50	68.77 ± 4.42	0.35
MA (mm)	64.41 ± 5.06	64.69 ± 4.92	0.80
EPL (%)	1.98 ± 2.91	2.20 ± 2.91	0.74
LY30 (%)	1.50 ± 2.51	2.01 ± 2.88	0.41
CI (1)	1.17 ± 1.52	1.25 ± 1.29	0.81
Post-OP			
R (min)	4.89 ± 0.87	4.76 ± 0.84	0.51
K (min)	1.13 ± 0.33	1.19 ± 0.31	0.46
α-angle (°)	72.89 ± 5.18	71.81 ± 4.64	0.34
MA (mm)	65.77 ± 6.25	66.11 ± 4.45	0.76
EPL (%)	2.30 ± 3.77	1.31 ± 1.41	0.06
LY30 (%)	1.94 ± 3.67	1.10 ± 1.19	0.09
CI (1)	2.17 ± 1.43	2.21 ± 1.12	0.91

TEG, thromboelastography; Pre-OP, pre-operative; R, reaction time, period to 2 mm amplitude; K, kinetics, period from 2 to 20 mm amplitude; MA, maximum amplitude; EPL, estimated percent lysis, estimated percent lysis within 30 min after MA; LY30, percentage of clot lysis 30 min after MA; CI, comprehensive coagulation index; Post-OP, post-operative.

**Table 4 jcm-12-01161-t004:** The sensitivity, specificity, predictive values and likelihood ratios at postoperative Anti-Xa activity values of 121.38 ng/mL.

aFXa (ng/mL)	Sensitivity (%)	Specificity (%)	PPV (%)	NPV (%)	LR (+)	LR (−)
121.38	57.14	70.83	43.24	80.95	1.96	0.61
180.12	14.29	100	100	0.75	-	0.86

aFXa, anti-factor Xa activity.

**Table 5 jcm-12-01161-t005:** Operation-related values of the ecchymosis group and non-ecchymosis group.

	Group A (n = 74)	Group B (n = 28)	*p* Value
Operation time (min)	101.26 ± 27.66	109.32 ± 36.13	0.23
Intraoperative blood loss (mL)	35.20 ± 32.84	28.93 ± 18.92	0.34
Hidden blood loss (mL)	218.64 ± 93.53	273.16 ± 92.19	0.01 *
Total blood loss (mL)	253.85 ± 89.05	302.09 ± 87.86	0.02 *

*, statistically significant difference in mean values between groups.

**Table 6 jcm-12-01161-t006:** Post-operative complications.

	Group A (n = 74)	Group B (n = 28)	OR (95% CI)	*p* Value
Total wound complications	3 (4.05%)	7 (25.00%)	7.89 (1.87, 32.21)	<0.01 *
Fat liquefaction or incision exudate	2 (4.05%)	6 (21.42%)	9.82 (1.85, 52.16)	<0.01 *
Poor healing of incision	1 (1.35%)	1 (3.57%)	2.70 (0.16, 44.76)	0.47
Wound infection	0	0	N/A	N/A
Asymptomatic DVT	3 (4.05%)	2 (7.14%)	1.82 (0.29, 11.52)	0.52
Distal only	3 (4.05%)	2 (7.14%)	1.82 (0.29, 11.52)	0.52
Proximal	0	0	N/A	N/A
Symptomatic VTE	0	0	N/A	N/A
All-cause inpatient mortality	0	0	N/A	N/A
Deep infection requiring return to surgery	0	0	N/A	N/A
Fatal bleeding	0	0	N/A	N/A
Major bleeding	0	0	N/A	N/A

N/A, not applicable; DVT, deep vein thrombosis; VTE, venous thromboembolism; *, statistically significant difference in mean values between groups.

**Table 7 jcm-12-01161-t007:** Regression analysis of potential risk factors for ecchymosis after TKA.

	Odds Ratio	95% Confidence Interval	*p* Value
Age (year)			0.55
Gender			0.34
BMI (kg/m^2^)			0.10
Total blood loss (mL)			0.81
Hidden blood loss (mL)			0.33
Postoperative EPL (%)			0.69
Postoperative LY30 (%)			0.82
Anti-factor Xa activity			
POD1	1.015	1.001–1.028	0.04
POD3	1.015	1.003–1.028	0.02

TKA, total knee arthroplasty; BMI, body mass index; EPL, estimated percent lysis within 30 min after maximum amplitude; LY30, percentage of clot lysis 30 min after maximum amplitude; POD, post-operative day.

## Data Availability

The data are available from the corresponding author on reasonable request.

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
