# Peer review of "The Role of Anti-Factor Xa Activity in the Management of Ecchymosis in Patients Receiving Rivaroxaban after Total Knee Arthroplasty"

_jcm, 2023, doi:10.3390/jcm12031161_

Round 1

Reviewer 1 Report

Dear authors,

thank you very much for giving me the opportunity to review this work.

Extensive perioperative blood loss and haematoma might not only restrict postoperative rehabilitation but might be also associated with severe complications. On the other hand, thrombembolic events are still a threat to all patients undergoing TKA. Perioperative monitoring parameters can help to find to ideal invidual dose of anticoagulants like rivaroxaban and therefore, your work is highly interesting.

In general, this work is well structured and well written. Nevertheless, i would like to raise some minor issues:

- Why did you chose a size of 3mm as cut-off for ecchymosis?

- At what p.o. day was the ecchymosis assessed/measured? 

- Is it possible to provide more pre- and postop. blood values, in particular the haemoglobin and thrombocytes?

Best regards

Author Response

Point 1: Thank you very much for giving me the opportunity to review this work. Extensive perioperative blood loss and haematoma might not only restrict postoperative rehabilitation but might be also associated with severe complications. On the other hand, thrombembolic events are still a threat to all patients undergoing TKA. Perioperative monitoring parameters can help to find to ideal invidual dose of anticoagulants like rivaroxaban and therefore, your work is highly interesting.

Response 1: Thank you for your positive comments and your efforts in reviewing our manuscript. Your comments are all valuable and very helpful for revising and improving our paper. We have studied the comments carefully and have made corrections accordingly. Below please find our responses to each of these comments.

Point 2: In general, this work is well structured and well written. Nevertheless, I would like to raise some minor issues: Why did you chose a size of 3mm as cut-off for ecchymosis?

Response 2: Thanks for your question. 3 mm is a cut-off value to distinguish ecchymosis from petechiae as per accepted guidelines, and it has been used in many published researches related to ecchymosis (Sarit Kalfon et al., J Pediatr Hematol Oncol, 2018; Ertan Yetkin et al., Clin Hemorheol Microcirc, 2018). Thank you for your reminding, and we have added the related references to the revised manuscript, which you could also find below.

The revised part (marked in red, Page 3, Line 117):

2.3. Perioperative management

Ecchymosis was defined as subcutaneous hemorrhagic lesions, larger than > 3mm, forming a flat, rounded or irregular, blue or purplish patch in this study. [26,27]”

Point 3: At what p.o. day was the ecchymosis assessed/measured?

Response 3: Thank you for your professional question. According to the published work, most of the patients developed ecchymosis within 2-3 days after TKA. (Wang et al., J Arthroplasty, 2018) However, to detect and record ecchymosis timely, we decided to monitor and assess the onset of ecchymosis in the enrolled patients daily from post-operative day 1 (POD1) to the discharge day. Now we have revised the related statement in the updated manuscript according to your kindly comment, which you could also find below.

The revised part (marked in red, Page 3, Line 118-119):

2.3. Perioperative management

Patients were monitored daily from POD1 to the discharge day by the researchers for the occurrence of ecchymosis after TKA, and then were divided into ecchymosis group and non-ecchymosis group later in the analyses.”

Point 4: Is it possible to provide more pre- and postop. blood values, in particular the haemoglobin and thrombocytes?

Response 4: Thanks for your professional and insightful question. Originally, we designed this study to figure out the relation between aFXa and ecchymosis after TKA, as well as the theoretical potential of aFXa in predicting ecchymosis. Thus, only aFXa was set as the primary outcome, and other parameters (including blood loss, PT and APTT) were regarded as secondary outcomes. As a result, we did not provide several blood values (including haemoglobin and thrombocytes) as you have noticed.

1) As ecchymosis was defined as subcutaneous hemorrhagic lesions, the blood loss was speculated to be more in patients with ecchymosis. Based on this context, we decided to record the blood loss of all enrolled patients. The Gross formula is the most widely used method to calculate blood loss, which is calculated based on the change of Hct. Hence, we just recorded the Hct values in this study.

2) Rivaroxaban is a novel oral anti-Factor Xa inhibitor, which was proved to have little influence towards platelet. Nonetheless, we still monitored its function through TEG, a comprehensive coagulation test providing information of both platelet and coagulation factors function. Maximum amplitude is one of the parameters of TEG, and reflects maximum strength of the clot, which is determined by the amount and function of platelets. Thus, we did not conduct a monitor of thrombocytes alone.

In addition, this study was conducted in 2019, and due to some objective reasons (our original medical record system was replaced by a novel one), it is less possible for us to collect more blood values now. Nevertheless, after discussion, our team come to a consensus that your comments are very insightful and valuable. We are going to collect more detailed blood values in the clinical trial being conducted in our center now, and provide a more comprehensive report towards blood and coagulation values in the future. Besides, we have added this issue as a limitation of our current study in the revised version, which you could also find below. Thank you again for your efforts in reviewing our manuscript.

The revised part (marked in red, Page 9, Line 262-264):

4. Discussion

There are some limitations to the present study. ... Fourthly, further prospective studies monitoring more coagulation parameters (such as platelet and coagulation factors) are needed to provide a more comprehensive view of the relationship of anti-factor Xa inhibitors, aFXa and coagulation function.”

Reviewer 2 Report

Review on: The role of anti-factor Xa activity in the management of ecchymosis in patients receiving rivaroxaban after total knee arthroplasty.

This study aimed to evaluate the efficacy of measuring anti-factor Xa activity in predicting ecchymosis after total knee arthroplasty. The study recruited 102 patients who received rivaroxaban and were divided into two groups: those with ecchymosis and those without. The study found that the aFXa levels in the ecchymosis group were significantly higher than those in the non-ecchymosis group, and that aFXa level was assessed as an independent risk factor for ecchymosis. Additionally, the study found that patients with aFXa levels above a certain threshold should be considered as high-risk population for postoperative ecchymosis and may require intense monitoring or dosage modification of anticoagulants. Overall, the study suggests that aFXa is a promising parameter for predicting ecchymosis after TKA.

2.2 Section, row 85, please replace received unilateral TKA with "underwent", same for the following exclusion criteria.

Consider moving Figure 1 to Methods section.

The Discussion section presents the key findings of the study, but it could be improved by providing a more detailed explanation of the significance of these findings in relation to previous research in the field of venous thromboembolic disease (VTE) prophylaxis during TKA. The authors should also provide a clearer explanation of the implications of the study's results for clinical practice and future research.

Overall, this is a great work of science, with clear presentation of methodology and results. My objections are minimal and I suggest Minor Revision.

Author Response

Point 1: This study aimed to evaluate the efficacy of measuring anti-factor Xa activity in predicting ecchymosis after total knee arthroplasty. The study recruited 102 patients who received rivaroxaban and were divided into two groups: those with ecchymosis and those without. The study found that the aFXa levels in the ecchymosis group were significantly higher than those in the non-ecchymosis group, and that aFXa level was assessed as an independent risk factor for ecchymosis. Additionally, the study found that patients with aFXa levels above a certain threshold should be considered as high-risk population for postoperative ecchymosis and may require intense monitoring or dosage modification of anticoagulants. Overall, the study suggests that aFXa is a promising parameter for predicting ecchymosis after TKA.

Response 1: Thank you for your efforts in reviewing our manuscript and your positive comments. We have studied the comments carefully and have made corrections accordingly. Below please find our responses to each of these comments.

Point 2: 2.2 Section, row 85, please replace received unilateral TKA with "underwent", same for the following exclusion criteria.

Response 2: Thanks for your kindly reminding. We have corrected all the parts you mentioned in the revised version, which you could also find below.

The revised parts (marked in red, Page 2, Line 85 & 88):

2.2. Participant

Patients were considered eligible if they 1) were 18 years of age or older; 2) received underwent unilateral primary TKA; 3) received rivaroxaban for VTE prophylaxis after TKA; 4) provided informed consent.

Patients were excluded if they 1) received underwent bilateral TKA or revision TKA; 2) were diagnosed with VTE or received vascular surgery in the last 6 months before admission; 3) were presented with comorbidities of severe hepatic dysfunction (Child-pugh C), renal diseases (creatinine clearance < 30 ml per minute) or other severe systemic diseases; 4) were pregnant or breast-feeding; 5) with any other contraindications of rivaroxaban.”

Point 3: Consider moving Figure 1 to Methods section.

Response 3: Thank you for your professional suggestion. We have moved Figure 1 to the Methods section as you have suggested, which you could also find below.

The revised parts (marked in red, Page 2-3, Line 94-96):

2.2. Participant

The flow chart of the study was illustrated in Figure 1.”

Point 4: The Discussion section presents the key findings of the study, but it could be improved by providing a more detailed explanation of the significance of these findings in relation to previous research in the field of venous thromboembolic disease (VTE) prophylaxis during TKA. The authors should also provide a clearer explanation of the implications of the study's results for clinical practice and future research.

Response 4: Thanks for your professional comments. The issues you raised are quite important to improve the quality of our manuscript, and we have discussed them carefully. We have added more detailed explanations along with the related references in the revised version. Besides, we have also added more statements towards how our results could help for clinical practice and future research. Please find the revised parts below. Thank you again for your professional review.

The revised parts (marked in red, Page 8-10, Line 216-219 & 223-224 & 245-247 & 262-264 & 269):

4. Discussion

Routine coagulation tests, such as PT, APTT and TEG, have been widely used in various surgical areas.[5,33] However, these parameters seemed to provide limited efficacy in monitoring and predicting ecchymosis. PT and APTT are the most frequently used clot-based assays, which are inexpensive and rapid, but they are less reliable in reflecting anti-factor Xa inhibitor exposure when compared with aFXa. [14] Similarly, no significant difference was detected in PT and APTT between the two groups in this study. In this context, it seems that chromogenic aFXa assay, which was applied originally for the determination of rivaroxaban plasma level, is a promising measurement for bleeding events monitoring in a variety of scenarios requiring anticoagulation.[34-36] Since the level of aFXa was proved to have superior linear relation with the anti-factor Xa inhibitor plasma concentration,[14] we hypothesized that patients with ecchymosis may have higher aFXa level than those without.[16-20]”

4. Discussion

In a previously published work, Wang et al. reported that the perioperative blood loss of patients with ecchymosis after TKA was more than those without.[5] Consistently, according to our results, patients with ecchymosis suffered significantly more blood loss and wound complications than those without (Table 5 & 6).”

4. Discussion

There are some limitations to the present study. ... Fourthly, further prospective studies monitoring more coagulation parameters (such as platelet and coagulation factors) are needed to provide a more comprehensive view of the relationship of anti-factor Xa inhibitors, aFXa and coagulation function.”

5. Conclusions

Patients with aFXa level > 121.38 ng/ml in clinic could be considered as high-risk groups for ecchymosis and may require dosage modification of anticoagulants.”

Point 5: Overall, this is a great work of science, with clear presentation of methodology and results. My objections are minimal and I suggest Minor Revision.

Response 5: Thank you for your positive comments and your efforts in reviewing our manuscript. Your comments are all valuable and very helpful for revising and improving our paper.
